# ABCFair: an Adaptable Benchmark approach for Comparing Fairness methods

**MaryBeth Defrance**
Ghent University
marybeth.defrance@ugent.be

**Maarten Buyl**
Ghent University
maarten.buyl@ugent.be

**Tijl De Bie**
Ghent University
tijl.debie@ugent.be

## Abstract

Numerous methods have been implemented that pursue fairness with respect to sensitive features by mitigating biases in machine learning. Yet, the problem settings that each method tackles vary significantly, including the stage of intervention, the composition of sensitive features, the fairness notion, and the distribution of the output. Even in binary classification, these subtle differences make it highly complicated to benchmark fairness methods, as their performance can strongly depend on exactly how the bias mitigation problem was originally framed.

Hence, we introduce ABCFair, a benchmark approach which allows adapting to the desiderata of the real-world problem setting, enabling proper comparability between methods for any use case. We apply ABCFair to a range of pre-, in-, and postprocessing methods on both large-scale, traditional datasets and on a dual label (biased and unbiased) dataset to sidestep the fairness-accuracy trade-off.

## 1 Introduction

Fairness has become a firmly established field in AI research as the study and mitigation of algorithmic bias. Thus, the range of methods that pursue AI fairness is now broad and varied [10, 41]. Many have been implemented in large toolkits such as *AIF360* [4], *Fairlearn* [48], *Aequitas* [45], or in libraries with a narrower focus like *Fair Fairness Benchmark* (FFB) [25], *error-parity* [13], and *fairret* [7].

So, which of these methods performs 'best'? Benchmarks in the past [5, 13, 18, 25, 29, 30, 37] tend to search for the best method *per dataset*. We argue such a benchmarking approach is of limited value in practice, as the real-world context of bias in AI systems imposes many subtle, yet significant, desiderata that a benchmark should align with before they can be properly compared.

Moreover, prior work commonly observes a trade-off: the more fair a model, the less accurate its predictions. This is unsurprising as fairness is typically pursued in settings where the training data *and the evaluation data* are both assumed to be biased [6]. Removing bias from predictions thus leads to degradation of this biased accuracy measure [49]. However, theoretical work has shown that this is not necessarily the case when evaluating on less biased data [16, 49].

**Contributions.** We formalize four types of desiderata that could arise in real-world classification problems. These include **1)** the stage where the method intervenes, **2)** the composition of sensitive groups, **3)** the exact definition of fairness, and **4)** the distribution that is expected from the model output. Figure 1 shows where these desiderata pose comparability challenges in a fairness pipeline.

We introduce *ABCFair*: an adaptable benchmark approach for comparing fairness methods. Through the use of three flexible components, the `Data`, `FairnessMethod`, and `Evaluator`, it can adapt to a range of desiderata imposed by the task. The approach is validated by benchmarking it on 10 methods, 7 fairness notions, 3 formats of sensitive features, and 2 output distribution formats.

38th Conference on Neural Information Processing Systems (NeurIPS 2024) Track on Datasets and Benchmarks.

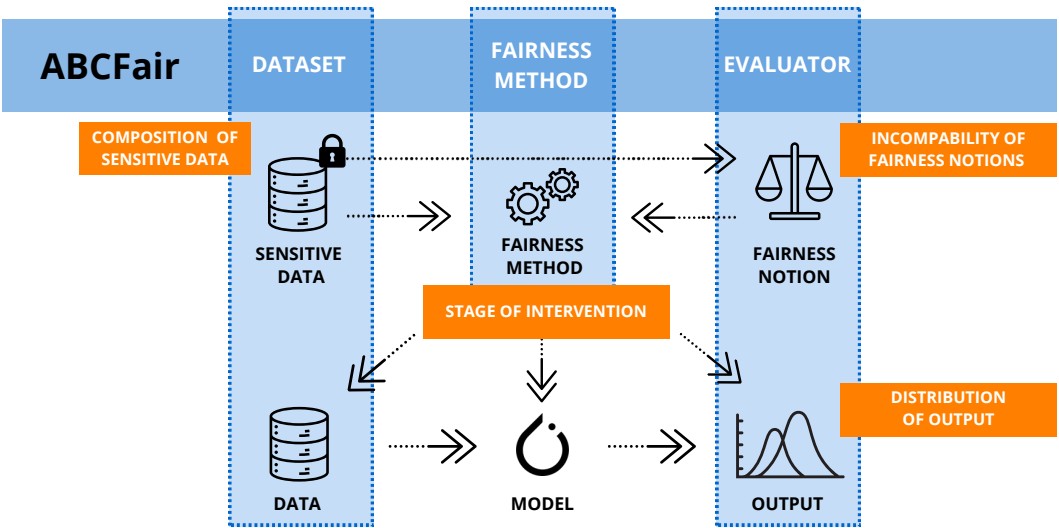

Figure 1: Structural overview of the ABCFair benchmark approach.

We further introduce the concept of evaluating on two types of datasets. A *dual* label dataset, which contains both biased and unbiased labels for evaluation [38]. Such datasets allow us to train a method with biased labels (simulating real-world settings), while still evaluating accuracy and fairness over less biased labels. We later refer to these less biased labels as unbiased to emphasize the difference with the traditionally biased labels. This allows us to challenge the notion of the fairness-accuracy trade-off [49]. We extend the analysis of the performance of bias mitigation methods for this dataset to focus on the fairness-accuracy trade-off [38]. To the best of our knowledge, we are the first to use a real-world dual label dataset in a benchmark.

However, dual label datasets are rare and often small. They present interesting findings, but a more robust dataset is needed for full evaluation. We therefore also introduce a more comprehensive evaluation method for large-scale traditional datasets, such as the folktables datasets [15]. Which is used in many benchmarks, even for topics other than bias mitigation method [11, 20, 19, 21, 24, 1].

All our code and results are available at `https://github.com/aida-ugent/abcfair`.

**Related Work.**  An early fairness benchmark was performed by Cardoso et al. [37]. They sampled synthetic datasets from Bayesian networks fitted to real-world data and measured model performance as a function of the bias in the dataset. Another early fairness benchmark was done by Friedler et al. [18], who were among the first to jointly consider multiple sensitive attributes and a wide range of fairness notions. Biswas and Rajan [5] take a more pragmatic approach by benchmarking different types of fairness methods applied to top-rated (and hence realistic) models from Kaggle.

Table 1: Quantitative comparison of fairness benchmarks. Fairness notions are only counted once per benchmark, even if they measure the violation of that notion in multiple ways.

| Benchmark | Dataset labels | | Methods | | | Fairness notions | Multiple sens. feat. |
|---|---|---|---|---|---|---|---|
| | Dual | Biased | Pre- | In- | Post- | | |
| L. Cardoso et al. [37] | ✓ | ✗ | 5 | 0 | 0 | 3 | ✗ |
| Friedler et al. [18] | ✗ | ✓ | 1 | 3 | 0 | 7 | ✓ |
| Biswas and Rajan [5] | ✗ | ✓ | 2 | 2 | 3 | 6 | ✓ |
| Fairea [29] | ✗ | ✓ | 3 | 2 | 3 | 2 | ✗ |
| Islam et al. [30] | ✗ | ✓ | 0 | 2 | 0 | 4 | ✓ |
| Fair Fairness Benchmark [25] | ✗ | ✓ | 0 | 6 | 0 | 6 | ✗ |
| Cruz and Hardt [13] | ✗ | ✓ | 2 | 3 | 1 | 3 | ✓ |
| ABCFair (ours) | ✓ | ✓ | 4 | 5 | 1 | 7 | ✓ |

Benchmarks tend to reveal an apparent trade-off between fairness and accuracy [49]. The Fairea [29] benchmark proposes a baseline for 'worst possible' trade-off that flips a model's outcomes completely at random in pursuit of a fairness constraint. Islam et al. [30] challenge whether a trade-off always occurs, as they empirically find that sensible hyperparameter tuning can already lead to improved fairness without loss in performance. More recently, the Fair Fairness Benchmark [25] (FFB) focusses on evaluating inprocessing methods and discuss their training stability. Yet, Cruz and Hardt [13] show that pre- and inprocessing methods anyway never achieve a better trade-off than simply applying group-specific thresholding to pareto-optimal classifiers, as such a postprocessing procedure is then pareto-optimal. They found this empirically holds for most pareto-optimal classifiers.

The novelty of our benchmarking approach is found in the comparability challenges we identify in Sec. 2, which we address with the *ABCFair* pipeline in Sec. 3. A quantitative comparison in Tab. 1 validates that our benchmark's coverage of methods and fairness notions is on par with prior work.

## 2 Comparability Challenges in Fairness Benchmarking

Out of all machine learning tasks, AI fairness research has been mainly focused on binary classification [41, 10]. This involves learning a classifier $h : \mathcal{X} \rightarrow \{0, 1\}$ where, for a random feature vector $X \in \mathcal{X}$, the prediction $h(X)$ is close to the random output label $Y \in \{0, 1\}$. Let $\mathcal{T}$ denote a model tuning procedure that pursues this goal for a training distribution $D$ over $X$ and $Y$, i.e. $h = \mathcal{T}(D)$, in a broad sense, including both fitting a neural net and meta-learning approaches [2].

In *fair* binary classification, the predictions $h(X)$ should be unbiased (enough) with respect to *sensitive* features $S \in \mathcal{S}$, such as gender or ethnicity. This, unfortunately, is already the greatest common denominator of desiderata that all fairness methods pursue. In what follows, we discuss the variation of this problem setting across methods, and how it leads to comparability challenges.

### 2.1 Stage of Intervention

Surveys partition fairness methods in three types [41]: *preprocessing*, *inprocessing*, and *postprocessing*. Preprocessing methods modify the training distribution $D$ to remove clear biases, such as undesired correlations between the sensitive features $S$ and the label $Y$ [32, 8]. Inprocessing methods modify model tuning $\mathcal{T}$, e.g. imposing a constraint during training [50]. Postprocessing methods only modify the output $h(X)$, e.g. separate classification thresholds for each demographic group [13].

The stage at which a fairness method intervenes is not a purely cosmetic distinction; it also constrains its applicability. Preprocessing requires access to the training data, which means third parties cannot use such methods on classifiers trained with private datasets. Conversely, dataset providers cannot make guarantees about the fairness of classifiers trained on their data by third parties (who may introduce new biases). Fairness methods typically also require access to the sensitive features $S$ [52], possibly entailing privacy concerns for e.g. postprocessing methods that require access to someone's sensitive information for every prediction. In practice, this is often infeasible [28].

Hence, fairness benchmarks cannot be blind to the stage in which a fairness method intervenes or when each method requires access to $S$.

### 2.2 Composition of Sensitive Features

There are many attributes that enjoy legal protection from discrimination [3]. Yet, the format in which these attributes are formalized as sensitive features $S \in \mathcal{S}$ can differ.

Many methods use a *binary* format [32, 36, 42, 26, 39], i.e. they assume there are only two groups to be considered: the advantaged and the disadvantaged. Formally, the domain of sensitive features $\mathcal{S} = 2$. Though this is indeed applicable to a binary encoding of gender, it is a coarse categorization in practice: the 'disadvantaged' group may contain some subgroups towards whom the bias is much less significant than others. Bias against the latter is then underestimated.

Fairness methods are more practical if they admit a *categorical* format [50, 2], i.e. an arbitrary amount of demographic categories $\mathcal{S} \in \mathbb{N}_{>0}$. Not only is this a better fit for non-binary demographics, it also allows for *intersections* of demographic groups, like black women [34], with respective domains

$\mathcal{S}_1$ and $\mathcal{S}_2$ to be encoded as categories in the product $\mathcal{S}_1 \times \mathcal{S}_2$. Of course, this quickly leads to a deterioration of statistical power when measuring bias against increasingly granular intersections.

A middle ground is to instead consider a *parallel* format, i.e. multiple axes of demographic attributes independently [7]. The domain of sensitive axes $\mathcal{S}_1$ and $\mathcal{S}_2$ remains its product $\mathcal{S}_1 \times \mathcal{S}_2$, but bias is only considered within a single $\mathcal{S}_k$ at a time. For example, unwanted behaviour within a model can be detected if it is i) biased against black people *or* ii) biased against women, but if it shows bias against the subgroup of black women.

Clearly, bias metrics depend on whether sensitive features $S$ are binary, categorical (possibly encoding intersections), or parallel. In our approach, we thus measure bias separately in each format. Where possible, we also configure the fairness method to optimize for it. Note that we apply binary binning to non-categorical sensitive attributes, like age, due to compatibility constraints for many methods.

## 2.3 Incompatibility of Fairness Notions

Mathematically, bias is a broad concept that could refer to a pattern in the data, the way the model works, or the model output [41]. Defining fairness over the output is by far most popular, and a wealth of empirical and theoretical results have made it clear that such a definition can take on a wide range of mathematical forms [47], which are often mutually incompatible [35, 14]. Observational studies have shown that humans are also not unanimous on the best definition [46]. Consequently, benchmarks of fairness methods keep track of multiple notions of fairness at the same time [18, 5, 30].

In fact, many methods can be configured to optimize for a specific definition out of several options. This again introduces comparability problems. Methods that *can* optimize a certain fairness definition will likely have an advantage of those that can not, making these methods only comparable for the notions where they intersect. This intersection is not always obvious either. The names of fairness notions can be ambiguous, e.g. with 'demographic parity' sometimes referred to as a subtype of 'group fairness' [41], as a synonym [47], or specified as 'strong demographic parity' [31]. Also, though a method may pursue demographic parity over the output, it may only actually be influencing intermediate representations [39] or even only the data [32], which may not perfectly align.

In our benchmark, we evaluate a range of well-known definitions of fairness, defined as parity between simple statistics computed over the output [7]. The idea is that practitioners can choose the statistic that fits with the real-world context of the task, and select the method that performs best while achieving the desired level of parity. We leave a comparison to other fairness paradigms like causal fairness [40] and multicalibration [27] to future work.

## 2.4 Distribution of the Output

A final, common challenge to comparability is the distribution of the output, i.e. whether fairness is evaluated over *hard*, binary decisions $Y \in \{0, 1\}$ or *soft* scores $R \in (0, 1)$. Fairness definitions are almost always defined in terms of hard decisions [47], so standard libraries like Fairlearn [48] are designed with hard decisions in mind. Hard decisions are then expected to be sampled from a Bernoulli distribution $R^Y (1 - R)^{1-Y}$ with the soft score $R$ as the parameter [2], or they can be obtained by applying a threshold to $R$ such as $Y = \mathbf{1}_{R \leq 0.5}$.

However, soft scores may be more desireable in some real-world cases, for example if the final decision $Y$ is deferred to a human decision-maker that can choose to consult $R$ [9]. This is especially likely in applications where decisions have a high impact on someone's life, which are precisely those applications that motivate pursuit of fairness in AI [6]. Whether fairness should be measured over hard or soft outputs therefore depends on how those outputs in the real-world context. To compare fairness methods for any realistic use case, one should thus also compare both hard and soft fairness.

Table 2: Overview of the datasets used in the benchmark. The reported number of features is after pre-processing the datasets.

| Dataset name | # Samples | # Features | Sensitive attributes |
|---|---|---|---|
| SchoolPerformance [38] | 856 | 20 | Sex, parent's education |
| ACSEmployment [15] | 3,236,107 | 34 | Sex, age, marital status, race, disability |
| ACSIncome [15] | 1,664,500 | 11 | Sex, age, marital status, race |
| ACSMobility [15] | 620,937 | 63 | Sex, age, race, disability |
| ACSPublicCoverage [15] | 1,138,289 | 113 | Sex, age, race |
| ACSTravelTime [15] | 1,466,648 | 1834 | Sex, age, race, disability |

## 3  *ABCFair*

To address the challenges raised in Sec. 2, we introduce a new benchmarking approach: *ABCFair* (Adaptable Benchmark for Comparing Fairness methods), that helps in both establishing qualitative comparability across methods and performing a quantitative evaluation.

The backbone structure of *ABCFair* follows a standard PyTorch-idiomatic [43] paradigm, such that it easily runs on a GPU and enables methods from the *fairret* and *FFB* frameworks (discussed below). *ABCFair* begins by splitting up, preprocessing and loading the data into a standardized `Dataset` class (subclass of a `torch.Dataset`). A neural net is then constructed and tuned in a standard train loop. Meanwhile, a `FairnessMethod` class is initialized that modifies the data, train loop, and/or model output. Finally, the output on a validation or test set is evaluated in an `Evaluator` class.

### 3.1  The `Dataset` Class

**Implementation.**  The `Dataset` class of *ABCFair* adapts the format of sensitive features to any of the variants identified in Sec. 2.2, configured as a hyperparameter. Similarly, it can be configured whether the sensitive features $S$ are included in the input features $X$ to the model (and not only available separately). In addition to $X$, $S$, and the labels $Y$, the `Dataset`'s `__getitem__` provides additional references to sensitive features in all other formats, such that the `Evaluator` class can be tracked for each format.

Tab. 2 provides details on the datasets available in the current implementation of the repository.

**Experiment details.**  First, we use the *SchoolPerformance* dataset [38], which is a dual label: it contains both biased and (less) unbiased labels. Biased labels were gathered by asking humans to predict school outcomes for the "Student Alcohol Consumption" dataset [12]. These labels are assumed to be far more biased than the actual outcomes, even though some bias likely persists. Nevertheless, the human annotations are used for training and the actual outcomes are only used for (far less biased) evaluation.

Second, we run *ABCFair* on the large-scale, real-world *American Census Survey* (ACS) datasets from `folktables` [15]. Five different binary classification tasks are defined for this data.

### 3.2  The `FairnessMethod` class

**Implementation.**  The `FairnessMethod` class is an abstract base class that can implement any preprocessing, inprocessing, and/or postprocessing functionality (see Sec. 2.1). The standard training loop for neural nets already occurs outside the `FairnessMethod`, so the *naive* baseline (where no fairness method is applied) is simply a naked inheritance of this class.

**Experiment details.**  Ten fairness methods are currently implemented in *ABCFair*. Tab .3 provides an overview of these methods and their properties with respect to the comparability challenges.

Table 3: Overview of the methods used in the benchmark and the desiderata they (can) address as identified in Sec. 2: their stage of intervention, sensitive feature format, the fairness notions it can enforce, and the type of output it optimizes. We also include their implementation's software package.

| Method | Stage | Sens. feat. format | | | Fairness notions | Output | Package |
|---|---|---|---|---|---|---|---|
| | | Bin. | Cat. | Para. | | | |
| Data Repairer [45] | Pre- | ✓ | ✓ | ✗ | pr | N/A | aequitas |
| Label Flipping [17] | Pre- | ✓ | ✓ | ✗ | N/A | N/A | aequitas |
| Prevalence Sampling [45] | Pre- | ✓ | ✓ | ✗ | N/A | N/A | aequitas |
| Learning Fair Repr. [51] | Pre- | ✓ | ✗ | ✗ | pr | Soft | aif360 |
| Fairret Norm [7] | In- | ✓ | ✓ | ✓ | pr, tpr, fpr, ppv, for, acc, F1-score | Soft | fairret |
| Fairret KL$_{proj}$ [7] | In- | ✓ | ✓ | ✓ | pr, tpr, fpr, ppv, for, acc, F1-score | Soft | fairret |
| LAFTR [39] | In- | ✓ | ✓ | ✗ | tpr + tnr, acc. | Soft | FFB |
| Prejudice Remover [33] | In- | ✓ | ✓ | ✗ | pr | Soft | FFB |
| Exponentiated Gradient [2] | In- | ✓ | ✓ | ✗ | pr, tpr, fpr, acc | Hard | fairlearn |
| Error Parity [13] | Post- | ✓ | ✓ | ✗ | pr, tpr, fpr, tpr+tnr | Hard | error-parity |

### 3.2.1  Preprocessing

**Implementation.**  These methods override the `FairnessMethod`'s `preprocess` function. It takes a `Dataset` in Sec. 3.1 as input and is expected to return a debiased `Dataset` as output.

**Experiment details.**  Four pre-processing methods are implemented, which take varying approaches. *Data Repairer* [45] and *Learning Fair Representations* [51] both aim to remove the correlation between the features *X* and the sensitive features *S*. Furthermore, *Prevalence Sampling* [45] mitigates sampling bias by under- or oversampling certain groups, while *Label Flipping* [17], directly changing the labels.

### 3.2.2  Inprocessing

**Implementation.**  These methods receive a tuning loop procedure as input in the `FairnessMethod`'s `inprocess`. The loop implements `fit` and `predict` functions as in the standard interface of `scikit-learn` [44], and can be fully replaced or overwritten.

**Experiment details.**  Five in-processing methods are present in *ABCFair*. The *Prejudice Remover* [33] and the *Fairret Norm* and KL$_{proj}$ [7] methods add an extra penalty to the loss that expresses the unfairness of the model. Similarly, *LAFTR* penalizes an adversarial network's performance in predicting sensitive features from its intermediate representations, in addition to an extra reconstruction loss. Finally, we wrap the *Exponentiated Gradient* method [2, 48] as a modification of the model tuning loop, which 'reduces' fair binary classification to a series of naive classification problems with weighted samples.

### 3.2.3  Postprocessing

**Implementation.**  These methods implement `FairnessMethod`'s `postprocess` function, which receives both the training `Dataset` and the tuned model's output as input. They should return a Python callable that performs postprocessed inference on new data samples (such a the test set).

**Experiment details.**  *Error Parity* [13] is the only postprocessing method currently implemented in *ABCFair*. It enjoys strong theoretical guarantees by creating a separate decision thresholds for each sensitive group such that they equalize a given fairness notion (up to a controllable bound).

### 3.3 The `Evaluator` class

**Implementation.** The `Evaluator` class monitors the predictions during training, validation, and testing. It computes the performance of the model for every prediction type, combination of sensitive attributes, fairness statistic, and performance measure.

**Experiment details.** Recall from Sec. 2.4 that we distinguish between hard predictions $Y \in \{0, 1\}$ and soft scores $R \in (0, 1)$. For soft scores, the `Evaluator` applies a threshold $Y = \mathbf{1}_{R \leq 0.5}$ to also obtain hard scores. Fairness is evaluated on both, but performance is only measured in terms of the accuracy for hard predictions and AUROC for soft scores.

Fairness is measured using the parity-based fairness definition in the `fairret` library. We refer to [7] for a full discussion, but summarize the methodology. First, the statistic $\gamma(q; h)$ for a relevant fairness notion is computed for every sensitive feature $q \in \{1, ..., |\mathcal{S}|\}$, over the model scores $h(X)$. We discuss results on two statistics $\gamma$ here, and report on five more in the Appendix.

First, *demographic parity* [47] requires equal *positive rates*:

$$\gamma(q; h) = \frac{\mathbb{E}\left[S_q h(X)\right]}{\mathbb{E}\left[S_q\right]}, \tag{1}$$

where $S$ is one-hot encoded for categorical features, i.e. $S_q = 1$ for samples in group $q$.

Second, *equalized opportunity* [26] requires parity in the *true positive rate*:

$$\gamma(q; h) = \frac{\mathbb{E}\left[S_q h(X) Y\right]}{\mathbb{E}\left[S_q Y\right]} \tag{2}$$

where we use $R$ instead of $Y$ if the model outputs soft scores.

The fairness *violation* is measured as

$$\max_q \left| \frac{\gamma(q; h)}{\overline{\gamma}(h)} - 1 \right|, \tag{3}$$

i.e. each $\gamma(q; h)$ is compared to the overall *mean statistic* $\overline{\gamma}(h)$, found by setting $S_q \equiv 1$.

This methodology works for all of the sensitive feature formats identified in Sec. 2.2. Fairness for intersections of sensitive attributes can be measures by simply one-hot encoding each intersection. For computing fairness with respect to multiple sensitive attributes in parallel, e.g. gender and ethnicity, we concatenate the encodings of gender and ethnicity. The overall fairness measure is then the maximal violation across both attributes. Hence, the fairness violation for multiple attributes is lower bounded by the violation for a single attribute. Also, the violation in the intersectional format is lower bounded by the violation for the parallel format, as the latter is an aggregate of the former.

## 4 Experiments

We innovate on the common approach to analyzing the accuracy-fairness trade-off in two ways. First, we benchmark on a dual label dataset, such that we can measure unbiased accuracy. Second, we benchmark on a large-scale dataset and configure a wide range of desiderata. We report test set results averaged over 5 random seeds (with different train/test splits) for a range of fairness 'strenghts'.

### 4.1 Performance comparison on bias and unbiased labels

Figure 2 shows the fairness-accuracy relation of several fairness methods. The top row shows the expected trade-off, as its accuracy is computed on test set labels that are i.i.d with the biased labels that the model was trained on. However, evaluating on the *unbiased* test set labels shown in the bottom row shows an accuracy-fairness *synergy*: an increase in fairness leads to an increase in accuracy. This practical result complements similar theoretical findings [16] and experiments on synthetic data [49].

**Key Finding 1: Methods that perform better on the traditional accuracy-fairness trade-off perform worse on unbiased labels.** This empirical result strongly motivates the creation of more dual label datasets, where a fairness method's true impact on accuracy can be properly assessed.

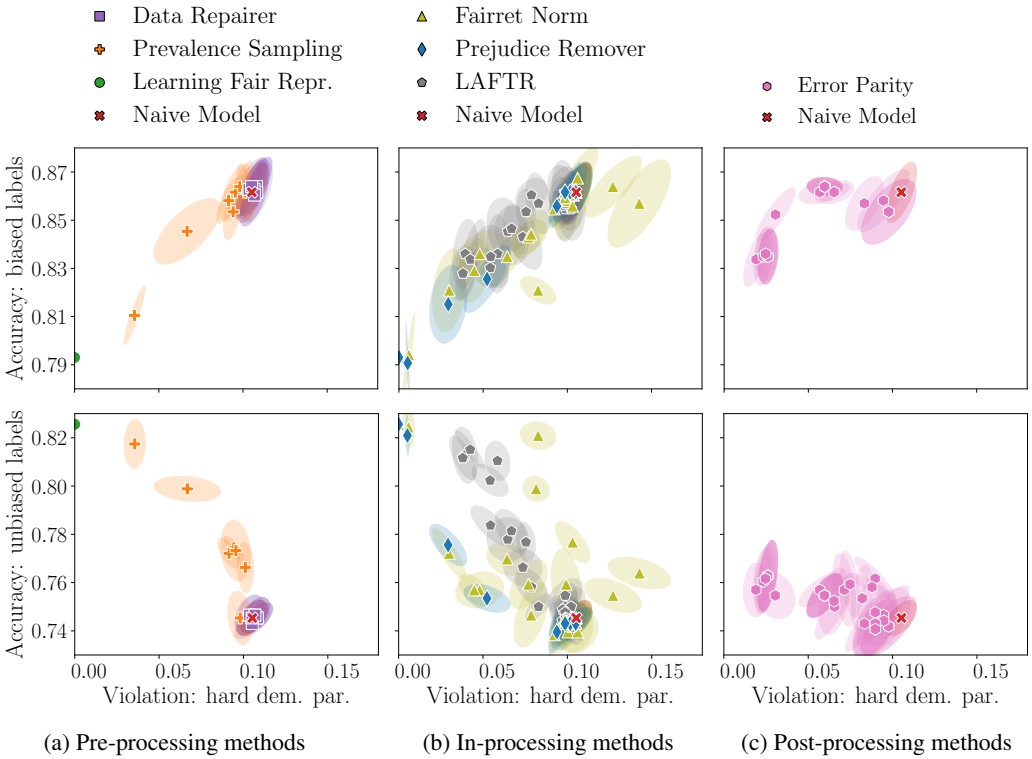

(a) Pre-processing methods     (b) In-processing methods     (c) Post-processing methods

Figure 2: Fairness-accuracy trade-offs on the SchoolPerformance dataset trained on biased labels. The top row is evaluated on biased labels and the bottom row on unbiased labels. Each marker is the mean test score over 5 random seeds, with a confidence ellipse (Appendix ) for 1 standard deviation.

## 4.2   Full result comparison on a real-world dataset

To validate the *ABCFair* approach on real-world data where only biased labels are available, we ran our full range of configurations on all five tasks of the ACS data in `folktables` [15]. Unlike the biased labels in the SchoolPerformance used in Sec. 4.1, there is no label bias in the ACS data. Other biases are still present in the data leading us to call the labels biased. We discuss a subset of the results here on the ACSPublicCoverage task and report all other results in the Appendix.

Though we do not have access to unbiased labels here, we still innovate on simply reporting a trade-off between (biased) accuracy and fairness. Instead, we start from the assumption that real-world applications of fairness will have a bound on the fairness violation in mind. If only biased labels are available for evaluation, then the most desirable method is whichever one that can achieve the best performance desired fairness violation 'level'. Hence, we report our results in Tables 4, 5, and 6 for three arbitrary fairness violation limits $k$ per configuration of sensitive feature composition (binary, intersectional, and parallel), fairness notion, and output format (as discussed in Sec. 3.3). $k$ denotes the maximal fairness violation a model may exhibit and still be consider fair.

As expected, the (biased) performance degrades for lower fairness violation bounds $k$ in all Tables.

**Key Finding 2: Preprocessing methods struggle to ever obtain very low fairness violations.** This is evident in Tab. 4, where the preprocessing methods (the first four columns) struggle to satisfy the lowest fairness violation thresholds $k$ even for very strong fairness strengths. *Learning Fair Repr.* manages it for non-binary sensitive features, but at a huge loss in performance.

**Key Finding 3: More granular sensitive features compositions lead to consistently larger fairness violations.** This was expected in Sec. 3.3, and we indeed see in Tab. 5, that the fairness violation for the intersectional sensitive features is lower bounded by the fairness violation for the parallel sensitive group configuration. Indeed, for the same violation levels $k = 0.10$ and $k = 0.20$ we see that the performance is higher when constrained on parallel groups compared to intersectional groups.

Table 4: The maximal **AUROC** and standard error in %, for the fairness strength where the **(soft) demographic parity** violation is smaller than $k$. A naive model achieves AUROC = 81.2% and fairness violation of Binary = 0.046, Intersectional = 0.48 and Parallel = 0.17.

| | $k$ | Data Repairer | Label Flipping | Learning Fair Repr. | Prevalence Sampling | Fairret Norm | Fairret $KL_{proj}$ | LAFTR | Prejudice Remover | Exponentiated Gradient | Error Parity |
|---|---|---|---|---|---|---|---|---|---|---|---|
| Binary | 0.01 | - | **81.2$_{\pm0.1}$** | - | - | **81.2$_{\pm0.0}$** | **81.1$_{\pm0.1}$** | 65.4$_{\pm3.6}$ | **81.2$_{\pm0.1}$** | 70.5$_{\pm0.3}$ | 69.5$_{\pm0.1}$ |
| | 0.03 | 80.2$_{\pm0.1}$ | **81.2$_{\pm0.1}$** | - | **81.2$_{\pm0.1}$** | **81.2$_{\pm0.0}$** | **81.2$_{\pm0.1}$** | 80.8$_{\pm0.0}$ | **81.2$_{\pm0.1}$** | 70.5$_{\pm0.1}$ | 69.5$_{\pm0.1}$ |
| | 0.05 | **81.2$_{\pm0.1}$** | **81.2$_{\pm0.1}$** | 63.0$_{\pm1.0}$ | **81.3$_{\pm0.1}$** | **81.2$_{\pm0.1}$** | **81.2$_{\pm0.1}$** | **81.3$_{\pm0.1}$** | **81.2$_{\pm0.1}$** | 71.1$_{\pm0.1}$ | 69.6$_{\pm0.1}$ |
| Inters. | 0.20 | - | - | 63.0$_{\pm1.0}$ | - | 79.3$_{\pm0.1}$ | **81.2$_{\pm0.1}$** | 68.8$_{\pm3.2}$ | 79.2$_{\pm0.1}$ | 66.3$_{\pm0.9}$ | 68.0$_{\pm0.1}$ |
| | 0.35 | 78.8$_{\pm0.1}$ | - | 63.0$_{\pm1.0}$ | 80.4$_{\pm0.1}$ | 80.8$_{\pm0.0}$ | **81.2$_{\pm0.1}$** | 80.1$_{\pm0.2}$ | 80.2$_{\pm0.1}$ | 70.5$_{\pm0.1}$ | 68.5$_{\pm0.1}$ |
| | 0.50 | **81.2$_{\pm0.1}$** | **81.2$_{\pm0.1}$** | 63.0$_{\pm1.0}$ | **81.3$_{\pm0.1}$** | **81.2$_{\pm0.1}$** | **81.2$_{\pm0.1}$** | **81.3$_{\pm0.1}$** | **81.2$_{\pm0.1}$** | 71.1$_{\pm0.1}$ | 68.9$_{\pm0.0}$ |
| Parallel | 0.08 | - | - | 63.0$_{\pm1.0}$ | - | 79.9$_{\pm0.0}$ | **81.2$_{\pm0.1}$** | 67.3$_{\pm2.5}$ | 79.2$_{\pm0.1}$ | 65.1$_{\pm1.2}$ | 68.0$_{\pm0.1}$ |
| | 0.12 | - | - | 63.0$_{\pm1.0}$ | - | 79.9$_{\pm0.0}$ | **81.2$_{\pm0.1}$** | 74.6$_{\pm1.6}$ | 79.2$_{\pm0.1}$ | 66.3$_{\pm0.9}$ | 68.0$_{\pm0.1}$ |
| | 0.16 | 79.7$_{\pm0.1}$ | - | 63.0$_{\pm1.0}$ | **81.2$_{\pm0.1}$** | **81.2$_{\pm0.0}$** | **81.2$_{\pm0.1}$** | 80.8$_{\pm0.1}$ | **81.2$_{\pm0.1}$** | 71.1$_{\pm0.1}$ | 68.5$_{\pm0.1}$ |

Table 5: The maximal **AUROC** in % and standard error, for the fairness strength where the **(soft) equal opportunity** violation is smaller than $k$. A naive model achieves AUROC = 81.2% and fairness violation of Binary = 0.021, Intersectional = 0.31 and Parallel = 0.22.

| | $k$ | Data Repairer | Label Flipping | Learning Fair Repr. | Prevalence Sampling | Fairret Norm | Fairret $KL_{proj}$ | LAFTR | Prejudice Remover | Exponentiated Gradient | Error parity |
|---|---|---|---|---|---|---|---|---|---|---|---|
| Binary | 5e-3 | - | **81.2$_{\pm0.1}$** | - | **81.2$_{\pm0.1}$** | **81.1$_{\pm0.1}$** | **81.1$_{\pm0.1}$** | 80.8$_{\pm0.0}$ | **81.2$_{\pm0.1}$** | 69.3$_{\pm0.0}$ | 69.6$_{\pm0.1}$ |
| | 0.01 | - | **81.2$_{\pm0.1}$** | - | **81.2$_{\pm0.1}$** | **81.2$_{\pm0.0}$** | **81.2$_{\pm0.1}$** | 80.8$_{\pm0.2}$ | **81.2$_{\pm0.1}$** | 69.3$_{\pm0.0}$ | 69.6$_{\pm0.1}$ |
| | 0.02 | 80.2$_{\pm0.1}$ | **81.2$_{\pm0.1}$** | - | **81.3$_{\pm0.1}$** | **81.2$_{\pm0.0}$** | **81.2$_{\pm0.1}$** | **81.2$_{\pm0.1}$** | **81.2$_{\pm0.1}$** | 71.1$_{\pm0.1}$ | 69.6$_{\pm0.1}$ |
| Inters. | 0.10 | - | - | - | - | 79.3$_{\pm0.1}$ | **81.1$_{\pm0.1}$** | 61.7$_{\pm2.7}$ | 78.3$_{\pm0.1}$ | 68.2$_{\pm0.6}$ | 68.3$_{\pm0.1}$ |
| | 0.20 | - | - | 63.0$_{\pm1.0}$ | - | 81.0$_{\pm0.0}$ | **81.2$_{\pm0.1}$** | 74.6$_{\pm1.6}$ | 79.2$_{\pm0.1}$ | 69.3$_{\pm0.6}$ | 69.2$_{\pm0.1}$ |
| | 0.30 | 80.2$_{\pm0.1}$ | 80.8$_{\pm0.1}$ | 63.0$_{\pm1.0}$ | **81.2$_{\pm0.1}$** | **81.2$_{\pm0.0}$** | **81.2$_{\pm0.1}$** | **81.2$_{\pm0.1}$** | **81.2$_{\pm0.1}$** | 71.1$_{\pm0.1}$ | 69.6$_{\pm0.1}$ |
| Parallel | 0.10 | - | - | - | - | 80.8$_{\pm0.0}$ | **81.2$_{\pm0.1}$** | 67.3$_{\pm2.5}$ | 78.7$_{\pm0.1}$ | 69.3$_{\pm0.6}$ | 68.8$_{\pm0.1}$ |
| | 0.15 | - | - | 63.0$_{\pm1.0}$ | - | 81.0$_{\pm0.0}$ | **81.2$_{\pm0.1}$** | 70.7$_{\pm2.9}$ | 79.2$_{\pm0.1}$ | 69.3$_{\pm0.6}$ | 69.2$_{\pm0.1}$ |
| | 0.20 | - | - | 63.0$_{\pm1.0}$ | - | **81.2$_{\pm0.0}$** | **81.2$_{\pm0.1}$** | 80.1$_{\pm0.2}$ | 80.2$_{\pm0.1}$ | 71.1$_{\pm0.1}$ | 69.6$_{\pm0.1}$ |

Table 6: The maximal **accuracy** in % and standard error, for the fairness strength where the **(hard) demographic parity** violation is smaller than $k$. A naive model achieves accuracy = 78.8% and fairness violation of Binary = 0.11, Intersectional = 0.31 and Parallel = 0.50.

| | k | Data Repairer | Label Flipping | Learning Fair Repr. | Prevalence Sampling | Fairret Norm | Fairret KL$_{proj}$ | LAFTR | Prejudice Remover | Exponentiated Gradient | Error parity |
|---|---|---|---|---|---|---|---|---|---|---|---|
| Binary | 0.01 | - | - | - | - | - | - | - | - | $49.6_{\pm 7.5}$ | $\mathbf{78.8_{\pm 0.1}}$ |
| | 0.05 | - | $\mathbf{78.7_{\pm 0.1}}$ | - | $\mathbf{78.8_{\pm 0.1}}$ | $\mathbf{78.8_{\pm 0.0}}$ | $78.5_{\pm 0.1}$ | - | $\mathbf{78.9_{\pm 0.1}}$ | $72.3_{\pm 0.2}$ | $\mathbf{78.8_{\pm 0.1}}$ |
| | 0.10 | $77.6_{\pm 0.1}$ | $\mathbf{78.8_{\pm 0.1}}$ | - | $\mathbf{78.8_{\pm 0.0}}$ | $\mathbf{78.9_{\pm 0.0}}$ | $\mathbf{78.8_{\pm 0.1}}$ | $\mathbf{78.9_{\pm 0.1}}$ | $\mathbf{78.9_{\pm 0.1}}$ | $72.3_{\pm 0.2}$ | $\mathbf{78.9_{\pm 0.1}}$ |
| Inters. | 0.50 | - | - | - | - | - | - | - | - | $72.3_{\pm 0.2}$ | $\mathbf{78.3_{\pm 0.0}}$ |
| | 0.75 | $77.1_{\pm 0.1}$ | - | - | - | - | - | - | - | $72.3_{\pm 0.2}$ | $\mathbf{78.8_{\pm 0.1}}$ |
| | 1.00 | $77.6_{\pm 0.1}$ | $\mathbf{78.8_{\pm 0.1}}$ | - | $\mathbf{78.7_{\pm 0.0}}$ | $78.6_{\pm 0.0}$ | $78.5_{\pm 0.1}$ | $78.1_{\pm 0.1}$ | - | $72.3_{\pm 0.2}$ | $\mathbf{78.8_{\pm 0.1}}$ |
| Parallel | 0.10 | - | - | - | - | - | - | - | - | $49.6_{\pm 7.5}$ | $\mathbf{77.5_{\pm 0.0}}$ |
| | 0.30 | - | - | - | - | - | - | - | - | $72.3_{\pm 0.2}$ | $\mathbf{78.3_{\pm 0.0}}$ |
| | 0.50 | $\mathbf{78.9_{\pm 0.1}}$ | $\mathbf{78.9_{\pm 0.1}}$ | - | $\mathbf{78.8_{\pm 0.0}}$ | $\mathbf{78.9_{\pm 0.1}}$ | $\mathbf{78.9_{\pm 0.1}}$ | $78.6_{\pm 0.1}$ | $\mathbf{78.9_{\pm 0.1}}$ | $72.3_{\pm 0.2}$ | $\mathbf{78.9_{\pm 0.1}}$ |

**Key Finding 4: Preprocessing methods optimize for specific a fairness notion more efficiently than inprocessing, but those improve all fairness notions at once by improving on one.** Recall from Tab. 3 that most methods can directly equalize positive rates (pursuing demographic parity), but fewer can do so for *true* positive rates (pursuing equalized opportunity). This is clear in Tab. 4 and 5, as even fewer strengths can be found for which preprocessing methods (which do not pursue equalized opportunity) satisfy the lower violation levels. Yet, though *LAFTR* and *Prejudice Remover* also cannot pursue equal opportunity directly, they do reach efficient trade-offs on this 'unintended' fairness notion as well, suggesting they target bias at a 'deeper' level.

**Key Finding 5: Whether the output consists of hard or soft scores has a significant impact on trade-offs.** Again, we already argued this comparability problem in Sec. 2.4. Here, *Exponentiated Gradient* and *Error Parity* are both methods that do not optimize with soft model scores in mind, causing them to incur significant drops in performance in Tab. 4 and 5 for even the highest $k$ bounds (which correspond with the unfairness of the naive model). The *Fairret* KL$_{proj}$ method incurs no performance loss for all levels of fairness violation, as it *does* optimize for soft scores. Conversely, *Error Parity* performs very well when its performance is measured as intended in Tab. 6: on hard, binary predictions.

## 5 Conclusions

We introduced *ABCFair*, a novel benchmarking approach focused on the challenges of comparing fairness methods with different desiderata. After an extensive discussion of these challenges, we provide a configurable pipeline designed to address each of these challenges. We finally provide guidance on benchmarking fairness methods accordingly, covering a wide range of configurations.

**Limitations.** Due to the breadth of desiderata configurations we cover, we only validate our approach on 10 methods and 6 (tabular) datasets. A more elaborate benchmark is needed with more types of fairness methods and datasets from different domains to corroborate our observations.

To fully populate the tables for all $k$ values, many fairness strengths need to be tried. A more efficient way of determining these $k$ values is left to future work.

This work only considers the design choices of method regarding between-group fairness. Recent work has called for also evaluation in-group fairness when benchmarking bias mitigation methods [22, 23].

Finally, we stress that our view of fairness was highly technical. Though such an approach can have practical value, it is inherently limited in truly addressing fairness as a socio-technical goal [6].

## Acknowledgments and Disclosure of Funding

This research was funded by the BOF of Ghent University (BOF20/IBF/117), the Flemish Government (AI Research Program), and the FWO (project no. G0F9816N, 3G042220, G073924N).

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

# A    Datasets

Our experiments were performed on two types of data: the dual label *SchoolPerformance* dataset, and the staple, large-scale *folktables* datasets. In this section, we provide more details on the preprocessing (not to be confused with *fairness* preprocessing) we performed for each dataset.

## A.1    SchoolPerformance

The SchoolPerformance dataset was created by Lenders and Calders [38]. This dataset is based on the "Student Alcohol Consumption"-dataset [12]. The *unbiased* labels are the labels of the original dataset and they indicate whether someone succeeded in their education. The biased labels are collected through human experiments, where human subjects are given some of the student's features and they note whether they think that student would succeed or not.

We used the sex and the education of the student's parents as the sensitive attributes for this dataset.

We removed all features that are other expressions of the labels (i.e. outcomes) and we removed the ID and name of the student from the dataset.

## A.2    Folktables

The following datasets are all part of the folktables [15] datasets. The following holds for all of the datasets: Age is encoded to a binary feature which encodes whether someone's age is higher or lower than the average value when calculating for intersectional groups. Smaller race categories are grouped in order to maintain statistical power.

### A.2.1    ACSPublicCoverage

The goal of the ACSPublicCoverage dataset is to predict whether someone is covered by public health insurance. Note that this is the only folktables dataset on which we report results in the main paper.

Sex, age, and rage are used as sensitive features for this datasets.

The features on ancestry and specific information of disability type are omitted in our use of the dataset. We deem these features as not relevant for this use case.

### A.2.2    ACSEmployment

The goal in the ACSEmployment dataset is to predict whether someone is employed or not.

Sex, age, marital status, race, and disability status are used as sensitive features.

We drop the column concerning relationship status as this is encoded in a less elaborate way in the marital status attribute.

### A.2.3    ACSIncome

The goal in the ACSIncome dataset is to predict whether someone earns more than $50.000 per year.

Sex, age, marital status, race and disability status are used as sensitive features.

We drop the column concerning relationship status as this is encoded in a less elaborate way in the marital status attribute.

### A.2.4 ACSMobility

The goal of the ACSMobility dataset is to predict whether someone has changed their address in the previous year.

Sex, age, race, and disability are the sensitive attributes.

The features on relationship status, ancestry, and specific disability type are omitted from the dataset.

### A.2.5 ACSTravelTime

The goal of the ACSTravelTime is to predict whether someone has to commute for longer than 20 minutes to work.

Sex, age, race, and disability are used as sensitive attributes.

Relationship status and employment status of parents are not included as features.

## B Experiment Setup

### B.1 Model Architecture and Training Hyperparameters

The underlying model in all experiments was a fully-connected neural net. All hyperparameters (including the number of hidden layers in the neural net) were chosen based on the performance on the validation set when applying no fairness method (the naive baseline). The resulting hidden layer sizes, learning rates, number of epochs, and batch sizes are reported in Table 7.

Table 7: Hidden layer size, learning rate, number of epochs, and batch size used per dataset.

|  | Hidden Layer Sizes | Learning rate | # Epochs | Batch size |
|---|---|---|---|---|
| SchoolPerformance | [64] | 0.001 | 80 | 64 |
| ACSPublicCoverage | [512,256,64,32] | 0.0001 | 40 | 2048 |
| ACSEmployment | [512,256, 64, 32] | 0.0001 | 40 | 2048 |
| ACSIncome | [512,256,64] | 0.0001 | 40 | 512 |
| ACSMobility | [512,256,64] | 0.0001 | 45 | 2048 |
| ACSTravelTime | [16,256,128,64] | 0.0001 | 20 | 1024 |

### B.2 Fairness Strengths

All fairness methods have a hyperparameter that regulates the strength of fairness. Unfortunately, the most suitable scales for these strengths varies significantly across methods. In Tab. 8, we detail which fairness strength we used for each method and the additional strengths that were used for the ACSPublicCoverage dataset. These additional strengths were selected manually to further populate Tables 4, 5, and 6 (in the main paper).

### B.3 Computational Resources

All experiments were conducted on an internal server equipped with a 12 Core Intel(R) Xeon(R) Gold processor and 256 GB of RAM. All experiments, including preliminary and failed experiments, cost approximately 800 hours per CPU.

This large computational cost results from the breath of the possible combinations of desiderata across a large set of methods.

Table 8: The standard and additional strengths used for each fairness method during training.

|  | Standard strengths | Additional strengths |
|---|---|---|
| Data Repairer | [0.1, 0.5, 0.8, 0.9, 1] | [1.3, 1.5, 2, 2.5, 3, 5] |
| Label Flipping | [0.001, 0.01, 0.03, 0.1, 0.3] | [0.5, 0.7, 1, 1.3, 1.5, 2] |
| Prevalence Sampling | [0.1, 0.5, 0.8, 0.9, 1] | [2, 3] |
| Learning Fair Repr. | [2, 5, 25, 50, 75] | [0.1, 0.5, 1, 10, 5000] |
| Fairret Norm | [0.001, 0.01, 0.1, 1, 3] | [0.0001, 0.5, 0.7, 5] |
| Fairret $KL_{proj}$ | [0.001, 0.01, 0.1, 1, 3] | [1e-05, 5e-05, 0.0001, 0.0005, 0.001] |
| LAFTR | [0.001, 0.01, 0.1, 0.3, 1] | [0.0001, 2, 3, 5, 7, 10] |
| Prejudice Remover | [0.001, 0.01, 0.1, 0.3, 1] | [1e-05, 0.0001, 0.0005, 2, 3, 5] |
| Exponentiated Gradient | [0.8, 0.9, 0.95, 0.99, 1] | [0.3, 0.5, 0.6, 0.7] |
| Error Parity | [0.005, 0.01, 0.05, 0.1, 0.3] | [1e-05, 5e-05, 0.0001, 0.0005, 0.001] |

## C   Additional Fairness Notions

In the main paper, we discuss the *demographic parity* (`dem_par`) and *equalized opportunity* (`eq_opp`) fairness notions. In our full benchmark, we consider 5 more [7]:

- *predictive equality* (`pred_eq`) requires *false positive rates* (fpr) $\gamma(k; h) = \frac{\mathbb{E}[S_k(1-h(X))]}{\mathbb{E}[S_k(1-Y)]}$ to be equal. It is a natural variant of equalized opportunity, but applied to negative labels.

- *predictive parity* (`pred_par`) requires *precisions* (ppv) $\gamma(k; h) = \frac{\mathbb{E}[S_k Y h(X)]}{\mathbb{E}[S_k h(X)]}$ to be equal.

- *false omission rate parity* (`forp`) requires *false omission rates* (for) $\gamma(k; h) = \frac{\mathbb{E}[S_k Y(1-h(X))]}{\mathbb{E}[S_k(1-h(X))]}$ to be equal. It is a natural variant of predictive parity, but applied to negative labels.

- *accuracy equality* (`acc_eq`) requires *accuracy* (acc) $\gamma(k; h) = \frac{\mathbb{E}[S_k(1-Y+(2Y-1)h(X))]}{\mathbb{E}[S_k]}$ to be equal.

- $F_1$-*score equality* (`f1_score_eq`) requires $F_1$-*scores* $\gamma(k; h) = \frac{\mathbb{E}[2 S_k Y h(X)]}{\mathbb{E}[S_k(Y-h(X))]}$ to be equal.

Note that the shorthand name for each notion corresponds to an option in Sec. D.1. Though we measure the violations of these notions, most methods are not designed to optimize for these lesser known notions. We refer to Tab. 3 in the main paper for an overview of which method can equalize which statistic (also shorthanded in the list above).

# D  Additional Results

In the main paper, we only report the results of one dataset for three possible configurations of desiderata. Many more configurations can reported for each of the datasets, as we evaluate on 6 datasets (+ 1 from the unbiased labels in SchoolPerformance), 7 fairness notions, and 2 output formats, bringing the total amount of Tables we can generate to 98. The amount of trade-off curves we can generate (like in Fig. 2) is again multiplied by the amount of sensitive feature formats (3), making 294 plots possible.

Including all these results would overly clutter the appendix. Hence, we make all our results available in our repo at `https://github.com/aida-ugent/abcfair` and provide a simple command line interface to generate the Tables and Figures as shown in the main paper.

## D.1  Performance Table Generation

The performance table allows for three configuration options: the dataset, the fairness notion with respect to which violation is measured, and the output format. Here, the $k$ values used to generate the table can either be edited into the script. If not, $k$ values will be automatically inferred from the fairness violation $k'$ of the naive baseline as the values $[k'/4, k'/2, k']$.

The command line options are:

```
--data_name [DATA_NAME]
    Name of the data set. Current options are
    ['ACSPublicCoverage', 'ACSEmployment', 'ACSIncome', 'ACSMobility',
    'ACSTravelTime', 'SchoolPerformanceBiased', 'SchoolPerformanceUnbiased']
--notion [NOTION]
    The fairness notion to be used. Current options are
    ['dem_par', 'eq_opp', 'forp', 'pred_par', 'acc_eq', 'f1_score_eq', 'pred_eq']
--output_type [OUTPUT_TYPE]
    The output type. Options are
    ['hard', 'soft']
```

## D.2  Trade-off Figure Generation

The accuracy-fairness trade-off figure has an additional configuration option: the sensitive feature format. To express uncertainty of the mean estimator of two-dimensional variables (the accuracy and the fairness violation), the plots show confidence ellipses, based on the methodology in [7] (Appendix D.4). The ellipse radii use the covariance matrix for the standard error.

The command line options are:

```
--data_name [DATA_NAME]
    Name of the data set. Current options are
    ['ACSPublicCoverage', 'ACSEmployment', 'ACSIncome', 'ACSMobility',
    'ACSTravelTime', 'SchoolPerformanceBiased', 'SchoolPerformanceUnbiased']
--notion [NOTION]
    The fairness notion to be used. Current options are
    ['dem_par', 'eq_opp', 'forp', 'pred_par', 'acc_eq', 'f1_score_eq', 'pred_eq']
--output_type [OUTPUT_TYPE]
    The output type. Options are
    ['hard', 'soft']
--sens_attr [SENS_ATTR]
    The sensitive attribute format. Current options are
    ['binary', 'intersectional', 'parallel']
```

