# OpenReview forum: "ABCFair: an Adaptable Benchmark approach for Comparing Fairness Methods"
_NeurIPS.cc/2024/Datasets_and_Benchmarks_Track — NeurIPS 2024 Track Datasets and Benchmarks Poster_

### Official Review · Reviewer_BMCx · 2024-07-24

**Rating:** 5
**Confidence:** 4
**Correctness:** Described in the review.
**Clarity:** Issues described in the review.

**Review:**

Despite being an interesting problem, this paper has serious issues that prevented this reviewer of recommending its acceptance.

The main problem of the paper is with clarity and formalism. It is really hard to understand what/how the authors have done.
Methodology isn't well described. There is an excess of details about implementation, but mathematically, important metrics and quantities are not well-defined. This issue is especially important for Section 3.3, which is not clear at all.

The description of results is not clear as well. The meaning of Tables 4, 5, and 6 are hard to parse and not properly defined. This reviewer could not see the points that the authors made in the "Key findings". Just as an example, the meaning of $k$ is unclear, given how it was defined in Section 3.3.

The main results are related to a single dataset, which is not appropriate for a benchmark paper.

Overall, this reviewer thinks that this is promising work. However, it needs considerable improvement (both in presentation and experiments) before being published in a top-tier conference (such as neurIPS benchmarks and datasets).

**Strengths:**

The paper approaches a relevant problem to the machine learning community.

**Additional Feedback:**

No additional feedback.

**Documentation:**

Authors provide a github with code and data.

**Ethics:**

No issues found.

**Limitations:**

Described in the review.

**Opportunities For Improvement:**

Described in the Review.

**Relation To Prior Work:**

Related work is discussed.

**Summary And Contributions:**

The paper proposes a methodology to compare fairness methods when dealing with binary classification problems. According to the authors, the contributions can be summarized as:
- categorizations of methods can intervene, composition of groups, definition of fairness, and distribution that is expexcted from the model output;
- experiments showing the utility of their framework.

---

> ### Author Rebuttal · Authors · 2024-08-16
>
> We would like to thank the Reviewer and are encouraged that they find our paper promising work. The Reviewer points out several notable faults which could lead to some confusion. However, per their helpful suggestions, these are addressed through a small amount of well-chosen, local changes.
>
> &nbsp;
>
> ### The use and meaning of $k$
> The confusion on the $k$ variable is fair as this term was overloaded. The $k$ used in the tables is completely distinct from the variable defined in Section 3.3 and was not noticeably defined in Section 4. We rectified this mistake by renaming the variable in Section 3.3. Furthermore, we extended and homogenized the description of $k$ in the Section 4: $k$ is an arbitrary ‘fairness violation limit’  , which dictates the maximal fairness violation that a model may exhibit on the test set, averaged over all random seeds.
> &nbsp;
>
> ### Representation of the results
> Tables 4, 5, and 6 are indeed a novel manner to report the performance of fairness methods. Prior work mostly focused on finding the ‘best’ fairness method overall, leading researchers to generate trade-off curves where one method ideally dominates all others. Although this is easy to generate and parse for researchers, these curves are not suited for evaluation in a critical region practitioners would be interested in. This critical region is determined by specific desiderata resulting from real-world use cases. Practitioners are interested in the best-performing method under the desiderata that are most suited to their needs. In our representation of the results, this can be read out directly by navigating to a single table row, e.g. depending on their fairness violation limit $k$, or the output format of their predictions.
>
> &nbsp;
>
> ### Clarity and formalism
> Upon reflecting on the structure of Section 3, we see how the Reviewer finds our methodology difficult to identify. Please note, however, that only the first paragraph of each subsection in Section 3 describes our implementation. The remainder of each subsection details our methodology. This approach felt natural as the methodology and implementation are heavily intertwined for a benchmark framework. Still, it significantly hurt the navigability of the paper. We now use named paragraphs to clarify where we discuss the data/method/evaluation implementation, and where we discuss the methodology.
>
> In particular, the fairness measures and fairness violations definitions were already presented in Section 3.3. However, we have wrapped them in definition environments to make them easier to find. The last paragraph of Section 3.3 now reuses the notations from Section 2.2 to improve its clarity. Please note that the remaining definitions of fairness notions can be found in Appendix C.
>
>
> &nbsp;
>
> ### Amount of datasets
>
> In the main body of the paper two datasets are discussed, the SchoolPerformance dataset and the ACSPublicCoverage dataset. These datasets are sufficient to support our key findings. Experiments on four additional datasets validate these findings. The combination of all possible fairness measures (both hard and soft) with each dataset would result in 98 tables like the three in the paper itself. To make these properly searchable, we decided to provide additional tables through a command line interface. A manual for this interface was already provided in Appendix D. Yet, we now also adjusted the main body of the paper to direct readers to this section of the appendix for additional results.

---

> > ### Comment · Reviewer_BMCx · 2024-08-19
> > **Clarifications OK**
> >
> > This review thinks that the clarifications helped to understand the paper, but not significantly. Score was improved from 3 to 5.

---

> > ### Author Response · Authors · 2024-08-20
> > **Response to Reviewer**
> >
> > We would like to thank the Reviewer for responding and are pleased that our rebuttal helped clear up some confusion.

---

### Official Review · Reviewer_bsKd · 2024-07-24
**Novel insights but missing some details and clarity could be improved**

**Rating:** 6
**Confidence:** 4
**Correctness:** Experimental setup seems sound.

**Review:**

The authors do an effective work of outlining the comparability challenges in fairness benchmarking, and detail desiderata for an effective benchmarking framework. Some claims used in the motivation are not entirely true, or are too bold to be said without actual evidence.

**Strengths:**

- The paper outlines the comparability challenges in fairness benchmarking, and details the desiderata for an effective fairness benchmarking framework;
- Comparisons in the literature often focus on one part of the ML pipeline (pre-processing the data, in-processing the training algorithm, or post-processing the output probabilities), this paper compares all three and attempts to draw insights that are generalizable to whole sets of approaches;
- The distinction between datasets with biased labels and unbiased labels is important for a proper evaluation of algorithmic fairness; And the results for the dual label dataset are quite interesting;

**Additional Feedback:**

N/A

**Clarity:**

The paper is generally well-structured but writing clarity could be improved (e.g., abstract should be mostly self-contained and clear).

**Documentation:**

Given that the major contribution is a python package for the ABCFair framework, it'd be helpful to have example notebooks and other documentation on how easily get started using it.

**Limitations:**

See section on “Opportunities for improvement”.

**Opportunities For Improvement:**

- Abstract is not entirely clear on the contributions of the paper, and not self-contained; I don’t follow what was meant by “…the greatest common denominator of a problem setting is small, significantly complicating benchmarking.”
- Line 22: “… fairness is typically pursued in settings where the training data and the evaluation data are both assumed to be biased” is not entirely true;
	- Most of the related work comparisons cited exactly in this paper use datasets based on census data (e.g., Adult, or the ACS/folktables datasets), which would be hard to argue that it doesn’t correspond to the true labels. Most claims of unfairness are not related to the labels and evaluation set being biased, but to biased outcomes resulting from training on true labels, reflecting the true inequality that exists in the world. If “biased labels” has some other meaning, please clarify in the paper and adapt the writing.
	- Furthermore, “removing bias from predictions thus leads to degradation of this accuracy measure” would also be true if both train and evaluation sets were unbiased.

_Minor:_
- Evaluating AUROC is interesting but not substantially meaningful for algorithmic fairness of binary decisions, where all metrics rely on a binary outcome. Differentiating between score/soft outputs and hard outputs is important, but it would be more meaningful in the context of uncertainty estimation and calibration rather than solely algorithmic fairness.
- Table 2: Number of features does not seem correct; e.g., ACSPublicCoverage has 19 features, not 113;

**Relation To Prior Work:**

The paper is properly framed with relation to prior work.

**Summary And Contributions:**

The paper presents a new benchmarking framework designed to address the comparability challenges in evaluating fairness methods with different approaches (pre/in/post-processing). Novel insights are drawn from a public dual label dataset (with biased labels and unbiased labels).

---

> ### Author Rebuttal · Authors · 2024-08-16
>
> We would like to thank the Reviewer for their in-depth review. We are encouraged that the Reviewer shares our opinion on the necessity of this type of work.
> &nbsp;
>
> ### Opportunities for improvement
> The term ‘greatest common denominator’ was meant to refer to the largest set of problem setting properties (the stage of intervention, composition of sensitive features, fairness notion, and distribution of the output) that are shared among a set of fairness methods. However, this term turned out to be too obscure. Instead, we rephrase the sentence on lines 5-6 to “Even in binary classification, these subtle differences make it highly complicated to benchmark fairness methods, as their performance can strongly depend on exactly how the bias mitigation problem was originally framed.”
>
> **Biased labels**
>
> The Reviewer correctly points out some lack of clarity regarding our usage of the term ‘biased labels’. Indeed, a distinction can be made between 1) labels that are themselves measured in a biased way and therefore incorrect, and 2) labels that are correct but where other types of biases affect the data such that it reflects an undesirable inequality. Situation 1) is referred to as *measurement bias* by Mehrabi et al. (2021), e.g. which occurs in recidivism prediction if we only know who was arrested and found guilty; a biased label of who actually recidivated. However, the Reviewer is correct to point out that census data like the ACS datasets fall under situation 2) because these labels are unbiased, correct measurements of actual outcomes, i.e. what we would want to predict if we did not care about fairness. Still, attributes like ‘income’ are subject to historical biases and other biases, which we may want to avoid reproducing in predictions. Ding et al. (2021), who introduced these datasets, indeed reported the presence of unwanted biases. In both situations, **we can treat the data labels as a biased version of what we want to be predicting instead**. Hence, we use the term ‘biased labels’ in all datasets.
>
> As argued (and observed) by Wick and Tristan (2019), the fairness-accuracy trade-off does *not* generally arise in situations where the labels are completely *‘unbiased’*. In fact, optimizing for fairness can help improve accuracy in such cases. In our experiments, where we use the relatively unknown SchoolPerformance dataset with both biased and ‘less biased’ labels, we make the same observation: optimizing for fairness improves the accuracy (when measured on less biased labels). Note that in the paper, we sometimes use the term ‘unbiased labels’ when referring to the ‘less biased’ labels to emphasize the juxtaposition between the two label collection methods.
>
> To clear up all confusion, we include the clarification above in lines 21-24 when discussing our notion of bias and lines 232-238 when discussing the properties of the ACS datasets with relation to biased labels.
>
> **Minor: differentiating between score/soft outputs and hard outputs less meaningful**
>
> We respectfully disagree with the Reviewer on the lack of meaning in considering fairness for soft predictions. As argued in lines 134-137 discussing the distribution of the output, there is a fairness use case for both hard and soft predictions. The Reviewer mentions that all metrics rely on binary predictions, however our paper employs a formula for fairness metrics suited for both hard and soft predictions as noted in Section 3.3. Due to page constraints, this section may have been overly condensed. We now more overtly clarify in this section that the distribution of the output can be taken into consideration for the fairness measures.
>
> **Minor: Table 2**
>
> The Reviewer was right to spot this discrepancy in Table 2, where we report the number of features *after* preprocessing the data, which includes one-hot-encoding its categorical features. We have updated the caption of Table 2 accordingly.
> &nbsp;
>
> Mehrabi N., Morstatter F., Saxena N., Lerman K., & Galstyan A. (2021). A Survey on Bias and Fairness in Machine Learning. ACM Comput. Surv. 54, 6, Article 115 (July 2022), 35 pages.
>
> Ding F., Hardt M., Miller J., & Schmidt L. (2024). Retiring adult: new datasets for fair machine learning. In Proceedings of the 35th International Conference on Neural Information Processing Systems (NIPS '21).

---

> > ### Comment · Reviewer_bsKd · 2024-08-19
> >
> > Thanks for the detailed response.
> >
> > I appreciate the distinction between the two types of "biased" labels. This distinction should definitely be part of the paper (perhaps in Sec. 3.1), as without it it's unclear what is meant by "unbiased" labels -- which is a central part of the paper. In fact, _SchoolPerformance_ seems to be in situation 1, and _ACS_ in situation 2. In this sense, the term "unbiased" is used for both types of bias at the same time.
> >
> > ---
> > > disagree with the Reviewer on the lack of meaning in considering fairness for soft predictions
> >
> > To clarify, risk scores are of course meaningful for algorithmic fairness, in the sense that they encode uncertainty in a prediction (as mentioned in the review). But one cannot find surprising the fact that methods optimized for soft risk scores perform better on a soft metric, and methods optimized for hard/discrete outputs perform better on the hard metric. In fact, there seems to be very little performance difference among soft-output methods (all achieve around 81.2 in Tables 4-5, and 78.8 in Table 6). This is precisely an example of the lack of "common denominator" mentioned in the paper, as these methods fulfill different desiderata and evaluation on a soft/hard metric can be misleading.
> >
> > As I see it, the main contribution of a paper proposing a fairer comparison of fairness methods would be showing how there is no "best" method: different methods fit problems with different desiderata. For each type of identified desiderata, the package could have a benchmark available to assess which method performs best on each specific scenario. Additionally, as one major contribution is the python package itself, the repository should have better usage examples and documentation (for example, I couldn't find any notebooks).

---

> > > ### Author Response · Authors · 2024-08-20
> > > **Response to Reviewer**
> > >
> > > Thanks for the additional, in-depth feedback!
> > >
> > > We are happy that we were able to clear up our use of the term 'biased labels'. We will include the distinction between both scenarios (and our motivation for treating them similarly) in the paper in Sec. 3.1 (as suggested) and in lines 33-40.
> > >
> > > > one cannot find surprising the fact that methods optimized for soft risk scores perform better on a soft metric, and methods optimized for hard/discrete outputs perform better on the hard metric.
> > >
> > > Indeed, once identified as an important property of how the problem setting is framed, it is unsurprising that fairness methods optimizing for soft/hard metrics perform better on soft/hard metrics as well. Not only do we agree on the importance of this distinction for evaluation, we highlight that no choice of output format is 'best' in all practical cases. This is in contrast to a fringe discussion in prior work (e.g. Padh et al. (2021) and Jiang et al. (2020)), where there has been a tendency to view hard fairness metrics as perfect, and soft metrics as approximate 'relaxations'. Instead, we see both as a valid design choice, and emphasize that the 'best' format depends on the problem setting.
> > >
> > > > the main contribution of a paper proposing a fairer comparison of fairness methods would be showing how there is no "best" method: different methods fit problems with different desiderata
> > >
> > > The Reviewer may agree that our paper indeed (quite trivially) shows this hypothesis holds: soft-output fairness methods (like Fairret and Prejudice Remover) perform worse at optimizing for hard fairness metrics, and vice versa for hard-output methods (like Error Parity). The Reviewer’s suggested hypothesis is closely related to the hypothesis we have for the paper: “Fairness methods primarily excel in problem settings for which they can be configured to optimize for the specific desiderata of those settings.” Of course, the Reviewer’s hypothesis follows from ours.
> > >
> > > > For each type of identified desiderata, the package could have a benchmark available to assess which method performs best on each specific scenario. Additionally, as one major contribution is the python package itself, the repository should have better usage examples and documentation (for example, I couldn't find any notebooks).
> > >
> > > We hope that our package can become an easy-to-use resource to improve benchmarking practices in fairness research. We have the idea to provide our results online in a more easily accessible format. One possibility would be to integrate the benchmarking framework with Hugging Face. We are currently working on improving the documentation of the library. We do not find a notebook a suitable manner of running the experiments as they are very computationally heavy and ideally ran in parallel. We will provide notebooks for visualizing the results as they were in the paper.
> > >
> > > Padh, Kirtan, et al. "Addressing fairness in classification with a model-agnostic multi-objective algorithm." Uncertainty in artificial intelligence. PMLR, 2021.
> > >
> > > Jiang, Ray, et al. "Wasserstein fair classification." Uncertainty in artificial intelligence. PMLR, 2020.

---

### Official Review · Reviewer_ZRyy · 2024-07-26
**Solid paper that makes progress on a highly-relevant topic in the area of benchmarking fair classifiers.**

**Rating:** 8
**Confidence:** 3

**Review:**

This is a solid paper that is a good fit for the NeurIPS Datasets and Benchmarks Track. The comparison problem is clearly described and motivated. The method they’ve developed appears to be sound. The experiments demonstrate the value of their work in better understanding fair classifiers. It is not high in the type of technical originality that might be expected in the NeurIPS Main Track, but for this track I don’t think that’s an issue at all. The experimental design and datasets used are in line with recent efforts to improve the benchmarking of fair classifiers.

Pros:

1. Addresses an important and open problem.

2. Relevant and useful to a broad group of people interested in fair classification.

3. The value of the method is demonstrated by the experiments and five “Key Findings” which are communicated well.

4. Well written.

5. Good fit for the Datasets and Benchmarks Track.

Cons:

1. It could benefit from a minor revision to address the issue raised in Relation To Prior Work.

2. Figure 2 is a little messy and there might be a more interpretable way to present it.

3. Some limitations mentioned in the Limitations section, but I think these are reasonable limitations for this type of work.

**Strengths:**

As stated in my review, this paper has many strengths. The problem is highly-relevant to both practitioners who may be considering which fairness interventions to use and researchers studying these interventions. In addition, helping to develop analysis that can capture a potential “accuracy-fairness synergy” is one of the most important directions in this research area.

**Additional Feedback:**

Line 101: “but if it shows bias” -> “but *not* if it shows bias”

Lines 137-138: Missing words in “Whether fairness should be measured over hard or soft outputs therefore depends on how those outputs in the real-world context.”

Line 236: Missing words in “…performance desired fairness violation ’level’.”

**Clarity:**

Yes. It communicates the background, methods, and results in a thorough and clear way. I think it would even be approachable for someone with limited background in this area.

**Correctness:**

Yes. The evaluation methods and experiment design are appropriate and performed correctly.

**Documentation:**

Yes. There is explanation and a link to code that would support reproducibility.

**Ethics:**

No. I do not have any ethical concerns with this work.

**Limitations:**

Yes. The authors have noted several limitations in a thoughtful way. These limitations are all reasonable/understandable and do not diminish the value of the present work. Most notably, this work is limited by considering fairness as somewhat of an abstract toy problem, removed from its social context. This is a serious limitation, but it is one shared by most technical work in this area, and this type of work still makes a valuable contribution despite the limitation.

**Opportunities For Improvement:**

I have noted some opportunities in other sections, especially Relation To Prior Work.

One other minor point is that the plots in Figure 2 are difficult to interpret, yet important. I wonder if the following would be better: A row of 7 smaller separate plots for (a) followed by a similar row of 7 plots for (b). I’d also like to see the performance of the naive model (the point at 0.089 and 86.2%/74.5%) visualized on each plot, maybe just as a small star.

**Relation To Prior Work:**

This is mostly clear and the authors do a good job of situating their work in the context of prior work.

However, there is some confusion/ambiguity around the relation to reference [31] Daphne Lenders and Toon Calders. Real-life Performance of Fairness Interventions - Introducing A New Benchmarking Dataset for Fair ML.

It would be helpful for the authors to more clearly separate the novel contributions of this work from [31] and recent followup work by Calders and others (perhaps the last two are so recent they can be excluded as concurrent):
- Favier, M., Calders, T., Pinxteren, S., & Meyer, J. (2023). How to be fair? a study of label and selection bias. Machine Learning.
- Goethals, S., Calders, T., & Martens, D. (2024). Beyond Accuracy-Fairness: Stop evaluating bias mitigation methods solely on between-group metrics. arXiv preprint.
- Goethals, S., & Calders, T. (2024). Reranking individuals: the effect of fair classification within-groups. arXiv.

The impressive dataset from [31] is crucial to some of the results here and there is overlap between the results of [31] and parts of this work (Figure 2, Key Finding 1, etc.). As I see it, a major new contribution here is how effectively this work uses the dataset from [31] to thoroughly compare very different methods across multiple fairness formulations and desiderata. I would like to see the authors discuss this more and clearly state what is entirely new versus what adds further evidence for what was already seen in [31].

**Summary And Contributions:**

This work approaches the challenge of comparing different fairness interventions that attempt to reduce bias and discrimination in binary classification. There are several issues that have made it difficult to compare fairness methods thus far. These include, but are not limited to: (1) differences in stage of intervention (pre-, in-, or postprocessing) between methods; (2) multiple ways to capture sensitive features and several theoretically incompatible fairness metrics; (3) the mismatch between hard scores in fairness definitions and soft scores in some practical applications; and (4) the difficulty of measuring unfairness without a truly fair ground truth in the dataset.

The authors presents a benchmarking approach, called ABCFair, that aims to overcome these challenges and facilitate better and easier comparison for different fairness methods as well as different types of fairness methods (e.g. preprocessing vs inprocessing). They test this method using recent datasets that were created or reconstructed with fairness benchmarking in mind and describe some findings from their tests.

---

> ### Author Rebuttal · Authors · 2024-08-16
>
> We would like to thank the Reviewer for their thoughtful review and highly actionable suggestions. We highly appreciate their positive assessment.
>
> &nbsp;
>
> ### Opportunities for improvement
> Figure 2 was indeed rather difficult to parse. Per your suggestion, we split up the figure into three rows: one for preprocessing, one for inprocessing, and one for postprocessing methods. This shows the performance each method more clearly, while keeping the necessary vertical space to a minimum. The new Figure 2 is attached to this message as a PDF and uses about 25% of the extra content page allotted in the camera-ready version. To have the results be completely clear, we also include plots of each method separately in the Appendix.
>
> &nbsp;
>
> ### Relation to prior work
> Favier et al. (2023) is indeed a useful reference to support our arguments made in Section 4.1 on the differences when evaluating with biased versus unbiased labels. We have included their work briefly in Sections 1 and 4.1.
>
> The other two papers (Goethals et al., Goethals and Calders) raise interesting points on the effect of bias mitigation methods on in-group fairness, which was not discussed in our paper. We added this limitation that our work focusses solely on between-group fairness and the caveats of leaving in-group fairness are out of scope.
>
> Per your comment, we also highlighted where our use of the dataset from Lenders and Calders (2023) is novel compared to the original paper. As suggested, the novelty is mainly found in the thoroughness of our methodology and our breadth of experiment configurations. We adjusted lines 33-37 in the introduction accordingly.
>
> &nbsp;
>
> ### Additional feedback
> We thank the Reviewer for noticing these writing mistakes and have corrected them accordingly.
>
> &nbsp;
>
> Favier, M., Calders, T., Pinxteren, S., & Meyer, J. (2023). How to be fair? a study of label and selection bias. Machine Learning.
>
> Goethals, S., Calders, T., & Martens, D. (2024). Beyond Accuracy-Fairness: Stop evaluating bias mitigation methods solely on between-group metrics. arXiv preprint.
>
> Goethals, S., & Calders, T. (2024). Reranking individuals: the effect of fair classification within-groups. arXiv.
>
> Lenders D. & Calders, T. (2023). Real-life Performance of Fairness Interventions - Introducing A New Benchmarking Dataset for Fair ML. In Proceedings of the 38th ACM/SIGAPP Symposium on Applied Computing (SAC '23). Association for Computing Machinery.

---

> > ### Comment · Reviewer_ZRyy · 2024-08-27
> > **Thanks for the changes/clarification**
> >
> > Thanks. The new Figure 2 looks good, and the pre/in/post breakdown is interesting to see.
> >
> > I would still like to see them split apart more in an eventual long version of the paper. One of the interesting features in the figure is the reversal of the fairness-accuracy trade-off. When they were all overlapping in one plot it was difficult to see how much of the trend was real versus a Simpson’s paradox or specific to certain methods because the colors were difficult to distinguish.

---

> > > ### Author Response · Authors · 2024-08-31
> > > **Thanks for the comment**
> > >
> > > We would Reviewer ZRyy for answering to our rebuttal and that they appreciate our new version of Figure 2. The new Figure 2 is indeed a significant improvement and we would like to thank them for their feedback.
> > >
> > > The split apart Figures will be available in the Appendix, but we will definitely keep their comment in mind.

---

### Decision · Program_Chairs · 2024-09-26

**Decision:**

Accept (Poster)

**Comment:**

This paper presents a methodology for the fair comparison of existing approaches to fair binary classification, with a focus on challenges related to: 1) different stages of intervention, 2) incompatible fairness metrics, 3) mismatch between hard and soft output scores, and 4) the absence of unbiased ground truth. During the rebuttal, the authors have well addressed the concerns by reviewers.

I am in favor of accepting this paper. However, I recommend that the authors include a more detailed comparison with existing benchmarks. For instance, the ACS datasets were originally proposed for fairness, and there are numerous subsequent benchmarks based on them, as seen in [1, 2].

[1] Subgroup Robustness Grows On Trees : An Empirical Baseline Investigation.
[2] On the Need for a Language Describing Distribution Shifts: Illustrations on Tabular Datasets.